# Origin of Species before Origin of Life: The Role of Speciation in Chemical Evolution

**DOI:** 10.3390/life11020154

**Published:** 2021-02-17

**Authors:** Tony Z. Jia, Melina Caudan, Irena Mamajanov

**Affiliations:** 1Earth-Life Science Institute, Tokyo Institute of Technology, 2-12-1-IE-1 Ookayama, Meguro-ku, Tokyo 152-8550, Japan; melina.caudan@elsi.jp; 2Blue Marble Space Institute of Science, 1001 4th Ave., Suite 3201, Seattle, WA 98154, USA

**Keywords:** speciation, species concept, origin of life, origin of species, compartmentalization, protocell

## Abstract

Speciation, an evolutionary process by which new species form, is ultimately responsible for the incredible biodiversity that we observe on Earth every day. Such biodiversity is one of the critical features which contributes to the survivability of biospheres and modern life. While speciation and biodiversity have been amply studied in organismic evolution and modern life, it has not yet been applied to a great extent to understanding the evolutionary dynamics of primitive life. In particular, one unanswered question is at what point in the history of life did speciation as a phenomenon emerge in the first place. Here, we discuss the mechanisms by which speciation could have occurred before the origins of life in the context of chemical evolution. Specifically, we discuss that primitive compartments formed before the emergence of the last universal common ancestor (LUCA) could have provided a mechanism by which primitive chemical systems underwent speciation. In particular, we introduce a variety of primitive compartment structures, and associated functions, that may have plausibly been present on early Earth, followed by examples of both discriminate and indiscriminate speciation affected by primitive modes of compartmentalization. Finally, we discuss modern technologies, in particular, droplet microfluidics, that can be applied to studying speciation phenomena in the laboratory over short timescales. We hope that this discussion highlights the current areas of need in further studies on primitive speciation phenomena while simultaneously proposing directions as important areas of study to the origins of life.

## 1. Introduction

Incredible biodiversity is one of the characteristics of life as we know it. Recent studies estimate a staggering total of 8.7 million species, with an error margin of plus or minus 1.3 million [1], and also gauge that 86% and 91% of all terrestrial and marine species, respectively, are yet to be discovered and cataloged. The importance of life’s species assemblage, or biodiversity, has long been recognized in agriculture and functional ecology [2,3,4,5]. In “On the Origin of Species” [6], Darwin described his observation that a plot of land sown with several distinct grass species was more productive than a similar grass planted plot with a single species. The positive correlation between biodiversity and productivity through complementary species interactions has since become a subject of numerous studies and theories [7,8]. Several empirical and theoretical studies connected biodiversity with the ecosystem’s stability [9]. According to the insurance hypothesis, biodiversity insures ecosystems against declines in their functioning due to environmental fluctuations. Thus, the existence of many species provides a better guarantee that some will remain functioning even if others fail [10]. Based on comparisons of mainland and island communities and other observational data, Elton [11] proposed that more diverse communities were more difficult to invade and, thus, resistant to invasion. Tilman [12] applied a resource competition model to predict that a novel species’ probability of invasion would decline sharply with increasing resident species numbers. Invaders would have to endure and thrive on the resources unconsumed by the established species. Unconsumed resources would decline with an increase in biodiversity; consequently, invader success would decline with diversity per the consumer–resource interactions model [13].

Biodiversity is thus an essential factor in modern life’s survival, stability, development, and growth. In this review, we explore such biodiversity principles as applied instead to chemical evolution leading to the origin of life. The different origin of life theories view the formation of prebiological species and diversification differently. First proposed by Wächtershäuser [14], the autotrophic origin theory envisioned life starting with carbon fixation powered by an energy flux created by geochemical processes at liquid–solid interfaces. A related theory, proposed by Russell and Hall [15], described gradual accumulation and complexification of organic matter occurred either on mineral surfaces (i.e., a two-dimensional life) or in networks of mineral pores. In the above models, the mineral surface or pore not only served as a catalyst but also enclosed the protolife and protected it from infinite dilution in the primeval ocean [14,16]. Barring Wächtershäuser’s discussion of the possibility of early phenotypic divergence at a mineral pre-cell level leading to the emergence of bacteria and archaea [17], the presence of co-existing pre-cell variants in the autotrophic theories is underdeveloped.

Unlike the autotrophic theories, a class of heterotrophic hypotheses instead sees the origin of life as a consequence of diverse cosmic and terrestrial chemical processes. In the 1920s, Oparin [18] and Haldane [19] independently proposed a similar scenario of an early reducing atmosphere influencing the synthesis of organic compounds and their subsequent assembly and evolution into entities capable of self-sustaining and replication. Oparin envisioned the formation of liquid–liquid separated colloidal particles, or coacervates, and their complexification and eventual evolution into modern cells. In “The Origin of Life on the Earth” [20], Oparin detailed the plausible prebiotic production of dissimilar coacervates, including simple two-component droplets, complex multiple-component particles, and “internally complex” coacervates, structured droplets resembling protoplasm. Oparin further proposed that the diverse coacervates, or different species, would undergo natural selection based on their stability in given environments. In recent years, the heterotrophic origin theories encompassed the “messy chemistry” hypotheses that explore organization mechanisms in vast multivariate chemical systems to understand the transition from complex non-life systems to biochemistry [21] and biochemical cycles [22]. Examples of the organization in complex chemistry include the formation of self-sustaining autocatalytic networks [23,24,25,26] that speciate by amplifying some chemical systems at the expense of others. In the context of the RNA world, a hypothetical stage in chemical evolution when RNA molecules supported both genotype (by being copied) and phenotype (by having catalytic activity), Joyce and Szostak [27] argued for the advantage of co-existence of multiple compartmentalized replicating RNA agglomerates. While RNA could, in principle, self-replicate and catalyze reactions in a homogeneous arrangement, breakdown into multiple delimited independent entities would prevent system crashes due to the evolution of parasites. A parasitic RNA sequence in solution could thus invade the entire system; however, in a compartmentalized system, the emergence of a parasite in one compartment would not affect the whole population [27].

In broad brushstrokes, there are two schools of thought regarding the origin of life. The first approach follows a single chemical system as it emerges into the first living organism, develops into the last common ancestor (LUCA) [28,29], and only subsequently diversifies. Alternatively, one might consider complex and diverse life before LUCA [30], which could have emerged from multiple origins [31]. If complex “messy” prebiotic chemistry and multiple origins are considered, the question of how prebiotic systems speciated and diversified becomes significant. In this review, we explore prebiotic speciation mechanisms with an emphasis on primitive compartmentalization. We briefly discuss the multiple definitions and concepts of biological species and the modes of biological speciation as well as the state of current knowledge of prebiotic compartment structure, properties, and function as speciation agents. Finally, we end with a discussion of laboratory strategies, namely microfluidic techniques, by which prebiotic and abiotic speciation processes can be initiated, affected, and studied. Due of the multidisciplinary nature of this paper, there may be some terms or phrases that are not completely familiar to all readers, and we have included a glossary of terms (Table 1). Through these discussions, given the fact that many prebiotically plausible chemical systems on early Earth were indeed “messy”, we hope to shed light on the importance of the further study on speciation in the context of primitive compartments. In particular, we believe that primitive compartments formed before the emergence of the last universal common ancestor (LUCA) could have provided a mechanism by which primitive chemical systems underwent speciation, and that primitive speciation events due to function of primitive compartments resulted in the origin of species (prebiotic species) preceding the origin of life. Only through simultaneously understanding the concept of species in biological context, coupled with further theoretical and laboratory simulation studies of primitive compartments, may we truly understand the process of primitive speciation.

## 2. Species Concepts and Definitions

Despite extensive efforts by biologists and philosophers ([31,32,33,34] and references therein), arriving at a more concrete definition and delimitation of species is surprisingly tricky. Nevertheless, species are primary units in virtually all biology branches, including but not limited to taxonomy, evolution, genetics, paleontology, and ecology [35]. The importance of species stems from their significance in the taxonomic framework utilized across life sciences.

The notion that an organism begets a similar organism has been accepted since classical antiquity [36]. The first definition of species, or morphologically similar organisms, is attributed to the 17th century’s John Ray, an English naturalist ([37] cited in [38]). In “Species Plantarum” [39], Carl Linnaeus then introduced a taxonomic system for plants based solely on the number and arrangement of the reproductive organs. In Linnaeus’ worldview deeply rooted in religious belief, the species were fixed and immutable of knowable numbers [40]. The fixity of species was subsequently challenged by Charles Darwin in “On the Origin of the Species” [6]. Based on the data collected on the Beagle expedition and on later observations, research, and correspondence, Darwin derived a theory of population evolution over generations through natural selection. Despite the title, Darwin was perplexed by the clustering of organisms into species. In Chapter 6, entitled “Difficulties of the Theory”, he noted, “Firstly, why, if species have descended from other species by insensibly fine gradations, do we not everywhere see innumerable transitional forms? Why is not all nature in confusion instead of the species being, as we see them, well defined?”

Departing from the above morphological definitions, in the early to mid-twentieth century, American biologists Ernst Mayr [41] and Theodosius Dobzhansky [42] developed a so-called biological concept of species (BCS). Coined in Mayr’s book [41], first published in 1942, the biological concept defines species as “groups of actually or potentially interbreeding natural populations, which are reproductively isolated from other such groups”. The concept suffered heavy criticism in two major areas: inapplicability toward asexual organisms [43,44] and impracticability in its rigid notion of reproductive isolation [45,46]. In recent years, many, 26 so far according to Wilkins [47], novel species concepts have been introduced. For example, the genetic similarity species concept is based on 16S rRNA gene sequencing [48] and DNA barcoding [49], the ecological species concept [50] refers to a set of organisms adapted to a given niche, and the evolutionary significant species concept [51] refers to a population of organisms considered distinct for purposes of conservation. This pluralistic approach toward species concepts [52], that directly contradicts Mayr’s BCS that assumes that species is a concrete phenomenon that cannot be arbitrarily recategorized [53], creates working definitions required in a particular research area for a specific purpose. Similarly, for the purposes of this review, we define prebiotic species as delimited chemical systems of similar makeup and properties. Prebiotic species might or might not have physical borders but must be identifiable as separate entities. The species would be composed of commensurate assortments of chemical compounds and would possess similar reactivity and physicochemical characteristics, such as solubility and colligative properties. These different chemical species would have emerged through differential selective processes, potentially resulting from interactions with primitive compartments, the possible mechanisms of which will be discussed in further detail in a later section.

## 3. Speciation in Biology

The process of speciation by natural selection was first illustrated by Darwin’s sketch (Figure 1) in his notebook nearly 20 years prior to the publication of “On the Origin of Species”. Darwin put forward a model whereby lineages diverge from their ancestors to become new species. Under the BCS, the formation of new species is driven by the evolution of reproductive barriers to the production of viable offspring. In allopatric speciation, reproductive barriers are introduced through geographic separation of previously interbreeding populations. The geographic separation allows the allopatric populations to diverge unimpeded by hybridizations. The allopatric populations inevitably then diverge due to different selective pressures, random genetic drift, and accumulation of different mutations in the separated gene pools [54,55]. Famous examples of allopatry include the unique species of finches [56] and tortoises [56,57] found on different Galapagos islands. Other less restrictive proposed modes of geographical speciation include peripatry [58], the formation of new species in peripheral populations, and parapatry [54], under which only partial geographical separation is afforded to diverging populations. When speciation occurs in the apparent absence of physical barriers, a sympatric mode of speciation is considered; the origin of *Rhagoletis pomonella*, the apple maggot, is often cited as an example of sympatric speciation. This species’ apple-feeding race appears to have spontaneously emerged from the hawthorn-feeding race in the early 19th century after apples were first introduced into North America. The apple-feeding race does not presently feed on hawthorns, and the surviving hawthorn-feeding race does not feed on apples [59,60]. Sympatric speciation can occur in heterogeneous habitats by adaptation to different ecological niches [61] or following the continuous resource model [62], under which sympatric groups adapt to utilizing different parts of resource distribution in their shared habitat.

Microorganisms are mostly incongruent with the BCS; the identification of microbial speciation mechanisms requires both asexual reproduction and varying modes and rates of genetic exchange to be taken into account. Asexually reproducing organisms are still subject to allopatric and sympatric niche adaptations [62,63]. Microbes further engage in horizontal gene transfer [64], an exchange of genetic material between organisms other than by vertical transmission from parent to offspring. Horizontal gene transfer involves either homologous recombination, the exchange of different alleles of homologous genes between similar organisms akin to sexual reproduction, or acquisition of brand new genes by nonhomologous recombination [65]. Thus, speciation in recombinogenic microorganisms has been described as fuzzy [66]. Nevertheless, analysis of environmental isolates and metagenomes showed that microbial communities consist of genotypic clusters of closely related organisms and that display cohesive environmental associations and dynamics that clearly distinguish them from other such clusters coexisting in the same samples [67].

As briefly discussed above, biological speciation is driven by different evolutionary processes. In the following sections, we discuss the speciation of prebiotic systems prompted by a set of abiotic processes constituting chemical evolution. In general, allopatry and adaptation to sympatric niches would be applicable to prebiotic chemistry; different chemical processes are expected to be dominant in different geological environments [68]. For example, Patel et al. demonstrated [69] the formation of a number of RNA, lipid, and peptide precursors in a cyanosulfidic protometabolic reaction variants network. The authors hypothesized that these variants would emerge in separate streams and pools and be controlled by dynamic co-flow schedules. Adam et al. described mechanisms by which radiolytic chemical reactions resulting in differential synthesis of prebiotic chemicals can be affected significantly by differences in physical environments and properties, which could emerge as a result of differential primitive geological environments [70]. Similarly, Vincent et al. also showed that chemical ecosystems with variable dilution and recursion rates (simulating differences in potential primitive geological environments) resulted in differential emergence of mutually catalytic systems over long periods of time [71]. Oparin discussed the formation of multifariously composed coacervate assemblies that were subject to natural selection through environmental perturbations [20]. The idea of evolving assemblies was later captured in the graded autocatalysis replication domain (GARD) theory, or composome model, a general kinetic model for homeostatic growth and fission of compositional assemblies with specific application toward lipids [72]. An example of compositional speciation has been demonstrated in oil droplets [73]. In this system, properties of oil droplets as a function of composition were probed via an automated evolutionary process. This study analyzed the movement, division, and vibration of combinatorially assembled four-component oil droplets in water to create fitness landscapes analogous to the genotype–phenotype correlations found in biological evolution.

However, probably the most considered mechanism of prebiotic speciation is compartmentalization, partitioning of a homogeneous system into broadly defined protocell compartments. One clear example of compartment speciation, observed by Adamala and Szostak, showed that fatty acid vesicles containing a seryl-histidine dipeptide grew while identical vesicles in the same solution, but without the dipeptide, shrunk [74]. This is likely due to seryl-histidine catalyzing the synthesis of N-acetyl-L-phenylalanine-leucinamide, which localizes to the vesicle membrane and decreases membrane fluidity, promoting fatty acid absorption from the surrounding solution (hence causing vesicle growth). This resulted in the emergence of two distinct vesicle species in the solution: those containing seryl-histidine that grew and those without seryl-histidine which shrunk. In the following sections, we discuss prebiotic compartmentalization and compartment-driven speciation in further detail.

## 4. Prebiotic Compartments

Compartments are a region of space containing a boundary where the internal components within a given compartment, in any state of matter (solid, gas, or liquid), are distinctly separated from external components. In prebiotic chemistry, these compartments can be of many sizes and structures (Figure 2) [75,76]. Such compartments can help to segregate and concentrate low-concentration reactants, helping to promote acceleration of essential primitive reactions [77]. Simultaneously, encapsulated components (such as essential prebiotic chemical building blocks) can avoid diffusion and dilution, which would have essentially rendered reactants inactive [78]. Such compartments could have also provided the ability to exclude certain molecules, such as those deleterious to essential chemical processes within the compartment (such as peptide inhibition of ribozyme-catalyzed RNA polymerization [79,80]) or waste [81], to the outside of the compartments. Finally, compartments can have very diverse structures and interiors, potentially offering a primitive system solvent diversity. This could have provided different reaction environments (outside of typical aqueous ocean or freshwater environments, which could result in degradation of certain biomolecules [77]) for primitive reactions to occur.

In order to accomplish the aforementioned functions on early Earth (such as segregation, concentration, exclusion, etc.), a primitive compartment would have required some type of boundary composed of components that could have existed and occurred in an early Earth environment. As modern cell boundaries are composed of phospholipid bilayer membranes, it has been postulated that a more primitive form of the modern cell may have also been composed of a lipid bilayer membrane system [82]. However, prebiotically plausible synthetic methods toward the complex multi-tailed phospholipids observed in modern biology appear to be challenging and, thus, simpler bilayer vesicles composed of single-chain amphiphiles, such as fatty acids, have been postulated to be a primitive model compartment [83,84,85]. A variety of such membrane-forming amphiphiles have indeed been found in meteorites [86] or as a result of other prebiotic synthetic processes [87]. Fatty acid vesicles have also shown to be assembled in geological conditions such as on mineral surfaces [88] or through processes such as dehydration/rehydration cycles in hot spring environments [89,90]; these dehydration/rehydration cycles could also have been necessary for polymerization of other prebiotically relevant polymers such as peptides [91,92], polyesters [93,94,95], or RNA [96]. While fatty acid vesicles have shown the ability to stably compartmentalize polymers (such as RNA) while still allowing smaller nutrients necessary for polymerization reactions (such as nucleotides) to freely diffuse into and out of the vesicle [97], pure (single-compound) fatty acid vesicles suffer in terms of stability to heat, magnesium ion concentration, and pH, often only being viable in a fairly narrow range of conditions. Increasing diversity of the fatty acid composition of vesicles, however, resulted in increased robustness and persistence to a number of environmental selection factors [98,99,100,101,102]. Further increasing diversity of the membrane composition by adding phospholipids to fatty acid bilayers imbues further selective advantages, for example by promoting greater compartment growth [103]. These studies suggest that a mixed phospholipid/fatty acid bilayer membrane vesicle may be an intermediate state between primitive single-chain amphiphile-based compartments and modern phospholipid-based cells. Compositional changes of compartments over time, such as increasing diversity of lipid vesicles, would have fit the GARD model, a general kinetic model for homeostatic growth and fission of compositional assemblies, specifically initially applied toward lipids [72]. In this model describing evolving assemblies, the compositional information of any system can change and evolve over time, as also demonstrated within oil droplets [73].

While modern cells are composed of such a lipid membrane bilayer, membraneless organelles generated from liquid–liquid phase separation are also abundant intracellularly and accomplish a variety of essential cellular functions [104]. As such, membraneless compartments have also been investigated in the context of primitive compartmentalization. Such systems can be of varying structural and functional complexity and could be produced from either associative (such as in complex coacervates [105,106,107]) or segregative (such as in polyethylene glycol (PEG) and dextran aqueous two-phase systems [78,108]) phase separation [109]. Other prebiotically plausible compartment systems have also been proposed, such as rock pores [110,111], polyester microdroplets [112,113], supercritical carbon dioxide droplets [114,115], or even combinations of different membraneless and membrane-bound systems [116,117,118,119,120].

A specialized case of a primitive compartment is a protocell, which is a compartment that eventually evolved into modern cells. A primitive protocellular compartment likely contained some type of replicating genetic material, primitive catalysts, and nutrients that could be incorporated into the synthesis or growth of the compartment, all encapsulated within a boundary system [83]. Some research has postulated that the first genetic material on early Earth was RNA, due to its ability to catalyze reactions, evolve, and replicate [27,121]. However, there still remain unanswered questions related to the plausibility of synthesis of RNA monomers, and efficiency of non-enzymatic polymerization of such monomers into functional and heritable RNA polymers in a prebiotic environment [81,122]. As such, proposals for the existence of pre-RNA [123,124] or other non-nucleic acid-based compositional modes of heredity [125,126] have also been explored. In any case, such a protocell would likely have required the encapsulation of one or more of these genetic materials to allow eventual protocell evolution. Additionally, incorporation of primitive catalysts within protocellular compartments may have been required to effect reactions promoting compartment growth, genetic evolution, and/or other essential reactions. These catalysts could have taken the form of RNA (as ribozymes) [127], short functional peptides [74,128,129], metal ions or metal-ion complexes (such as iron–sulfur clusters) [130,131], or even mineral particles [108,132,133].

A detailed discussion regarding assembly and structure of specific compartment systems is beyond the scope of this review, and a large number of resources on this topic exist, which we encourage the reader to refer to: [109,134,135,136,137,138,139,140,141,142,143].

## 5. Prebiotic Compartment Speciation

Compartmentalization offers intuitive mechanisms for affecting prebiotic speciation, such as partition of a complex prebiotic chemical system into disparate delimited subsets or exclusion or concentration of certain elements within a compartment, differentially affecting the rates of possible chemical reactions or processes. Herein, we set out to classify and catalog examples of compartment-assisted speciation mechanisms and outcomes (Figure 3). Generally speaking, two types of compartmentalization, discriminate (Figure 3B,C) and indiscriminate (Figure 3D,E) partitioning, prompt speciation. In discriminate speciation, specific subsets of molecules are placed within specific compartments, whereas in the indiscriminate process, random samplings of the initial systems are partitioned within compartments arbitrarily.

The simplest example of discriminate speciation involves sequestration of a specific set of molecules within a compartment, depleting the concentration of these molecules in the surrounding solution, forming a chemically different “inside” and “outside” species (Figure 3B). Studies have shown that amino acids and dipeptides within coacervate droplets can withstand normally degradative UV radiation [144], while ribozyme activity within a coacervate droplet can be enhanced [79], both scenarios of which could have resulted in synthesis of further biological building blocks within the compartment. In both aforementioned examples of emergent functions within systems as a result of coacervate compartmentalization (i.e., protection of amino acid/dipeptide UV degradation and enhancement of ribozyme activity within coacervates), a significant divergence in the structure (and potentially function) of entities within (i.e., undegraded dipeptides or ribozymes with enhanced activity) vs. on the outside of (i.e., degraded dipeptides or ribozymes without enhanced activity) the compartment would have resulted.

Further divergence would occur if more than one type of selective compartment is present (Figure 3B). The encapsulation (and depletion) of the different sets of chemicals would affect the possible chemical reactions amongst the prebiotic species, resulting in continuing divergence (Figure 3C). Selective pressures leading to speciation can also come from the degree of permeability of compartment boundaries, such as in lipid vesicles [97], which can stably compartmentalize long polymers but still allow small molecules to diffuse across the membrane. In fact, depending on the level of permeability, molecules such as essential nutrients or reaction waste (which could be potentially inhibitory to essential primitive reactions [81]) can circulate in and out of the compartment, while products of primitive biosynthesis could be retained inside. Furthermore, if a compartment system were sufficiently impermeable to polymeric or larger materials, then complex networks of catalyzed reactions (i.e., of long polymers or macromolecules) could develop within, but not on the outside of, a compartment [145]. Crowding within compartments [146], possibly caused by compartmentalization of high concentrations of solute, could also be one function provided by compartments that contributed to speciation by affecting the thermodynamics and the kinetics of reactions. For example, it has been shown that RNA ribozymes function more efficiently in a PEG-crowded environment [147], while RNA evolution has also been studied in a similar context [148]. Thus, the existence and degree of molecular crowding could have had a major role on the selectivity of molecule conformations, reactions, and products available within a compartment vs. outside of a compartment, potentially leading to further speciation.

A compartment’s ability to grow, replicate, and divide could also have contributed to speciation. One such example, mentioned previously, is the emergence of distinct fatty vesicle species that grow or shrink based on incorporation of a catalytic seryl-histidine dipeptide [74]. Inside a primitive compartment, a complex network of replicative autocatalytic reactions could also potentially emerge [149,150] and would survive by being fed by a constant input of nutrients from the outside. In some cases, some molecules or systems of molecules would be favored, i.e., thermodynamically, kinetically, by possible self-replication, etc., and be replicated with an increasing accuracy and efficiency (whether due to being in a more favorable environment within the compartment interior, or due to specific characteristics of the compartment itself which promote such reactions). Step-by-step, such a compartment which could grow could result in duplication of its components and, at some point, these components could effect the division of the compartment itself into two or more independent systems [140]. Progressively, this so-called “Darwinian selection” could expand from the molecular level to the compartment level. Compartments then with the most readily copied molecules in its interior, and also with the most functional catalytic molecules, would have a competitive advantage over other compartments. Gradually, the population of the selected species will grow and get transformed in turn, eventually resulting in compartment-level speciation [74,151].

In the indiscriminate scenario, the molecules in the initial homogeneous system are uniformly distributed within compartments (Figure 3D). The speciation ensues if the compartmentalized systems are subjected to different conditions or if the compartment itself affects the chemistry of the encapsulated molecules (Figure 3E). Alternatively, delimited chaotic systems [152,153] can lead to different outcomes akin to random genetic drift.

In fact, it has been demonstrated that indiscriminate compartmentalization, even transiently [154], of a pool of parasitic and non-parasitic replicating individuals is required to prevent the accumulation of parasitic individuals in a given primitive population, which would have resulted in system death and inhibited long-term evolution [155]. In these studies using replicating genetic systems, researchers discovered that in the absence of any compartments, parasitic agents which do not assist in replication of the system (the desirable characteristic needed for genetic evolution and, subsequently, survival) emerged as the dominant species. This resulted in an overconsumption of the available resources of the system by the parasites without any output that assisted in the survival of the system; in fact, in a regime of limited resources (the external perturbation), such an event would result in rapid extinction. However, in the presence of liposome compartments, the species that could result in system replication could be segregated from the parasitic species. This “speciation” event in essence resulted in two species: compartments containing replicating systems being able to evolve so that they could survive in the current conditions, and compartments containing parasites could not survive in the long-term. Speciation of generalized primitive compartments containing evolvable and replicable genetic components (with parasites), similar to those demonstrated experimentally above, has also been modeled through cellular automata [156,157]. Thus, to complement such computational research, ongoing research has been focused on demonstrating evolution and speciation of genetic molecule-encapsulating compartments of a variety of prebiotically relevant constructions such as lipid vesicles, oil-in-water droplets, or even more complex life-like artificial cells [102,138,140,158,159,160]. In fact, chemical evolution and speciation has even been recently described in far-from-equilibrium conditions even in mineral environments [161], themselves proposed to be one of the earliest compartment forms on early Earth [162], suggesting that chemical speciation likely had been occurring far before the emergence of the first precursors to modern compartments (i.e., a protocell).

As mentioned above, in addition to lipid vesicles, membraneless droplets generated from liquid–liquid phase separation [104,163], including polyester microdroplets [112,113], peptide/nucleotide coacervates [119], DNA droplet-based compartments [164], DNA liquid crystal coacervates [118,120], and aqueous two-phase systems [108], have been investigated in more detail in the context of primitive compartments as well. The overwhelming literature of a breadth of primitive compartments shows that the compartments themselves were also likely to have been extremely compositionally diverse and co-existed on early Earth, demonstrating properties that were not all necessarily compatible. It is also likely that various chemical systems co-existed during the chemical evolution leading to the first cells, potentially subjecting such a “messy” chemical milieu to compartmentalization (whether discriminate or indiscriminate). This would have produced a wide variety of interactions between various reactants in a primitive compartment and interactions between various reactants and the compartment itself. Such reaction diversity coupled with the potential diversity of the interior solvent and other external environmental conditions (such as temperature, pressure, salinity, or pH, assuming that compartment systems could survive such conditions [98,165]) could result in a nearly unlimited number of combinations of compartments containing chemical systems, leading to a larger probability of the emergence of a number of different species.

Ultimately, it is believed that the end-result of primitive evolution was the assembly and emergence of a cellular entity known as LUCA, which is postulated to be the precursor of all modern life [166,167]. To reach that point, a primitive compartment system likely underwent a significant amount of speciation events. Yet, it is generally believed that there is only one LUCA (outside of the “progenote model” [168]), and so accompanying the number of speciation events would likely have been a number of extinction events (or perhaps convergent evolution events) as well. In particular, LUCA would have emerged due to a number of properties which were favorable for metabolism, evolution, persistence, and other important life-like properties, while the extinct species likely possessed only few or even none of these essential properties, resulting in their extinction (i.e., selective evolution). Once LUCA emerged, subsequent evolution of LUCA likely resulted in more widespread speciation into precursors of the various kingdoms of living organisms and, eventually, into the various branches of the tree of life that we can observe today [169]. Yet, all (or most) of the new species which emerged from LUCA likely still contained the important properties that allowed such organisms to remain living.

## 6. Droplet-Based Microfluidics: Speciation in Action

Biological speciation is studied through a combination of fossil record, ecological surveys, and experiments [54]. Since observational data is not readily available for prebiotic chemistry in general and prebiotic speciation in particular, methods executable in laboratories over short timescales to observe such phenomenon must be devised. Herein, we survey droplet microfluidics, a technique that exemplifies the process of speciation in non-biological systems. This method involves in situ formation of many varied droplet compartments and can effectively represent and present different reaction or incubation conditions that primitive compartments could have potentially been subjected to on early Earth. Droplet-based devices manipulate discrete miniature volumes of immiscible fluids with low Reynolds numbers in laminar flow regimes [170]. This type of microfluidics allows for the generation and control of multiple microdroplets of varying composition or carrying different reactants to simultaneously create microenvironments ideally suited for many chemical, prebiotic, and biological applications and processes.

Such microfluidic devices generally generate microdroplets in passive or active modes. In passive systems, one immiscible fluid (dispersed fluid) is introduced into another (continuous fluid) through commingling capillary flows [171,172]. Active droplet generation methods rely on electric, magnetic, centrifugal, optical, or piezoelectric forces as well as surface acoustic waves to generate the droplets. The active techniques are designed to stabilize droplet generation and deliver a higher level of control over the production rate and droplet volume compared to the passive techniques [173].

One way of generating droplets with different compositions and properties within one experiment is to continuously vary the chemical concentrations of droplet components. Concentration gradients within microfluidics devices can be achieved by coupling the droplet generation systems with a network of microchannels with programmable flow (e.g., a Christmas tree network [174]) [175]. Alternatively, differential composition of droplets could be achieved through a controlled droplet coalescence. In one example, the disparate conditions within the microfluidic droplets were created by forming a local temperature differential across the droplet generation [176]. For such experiments and analyses, the reagents and analytes can be introduced into droplets via a co-flow before the droplet formation [177], by a fusion of droplets with a different composition [178], or by injection into existing droplets [179,180].

Multiple droplets of unique composition, however, would require individual means of transport and analysis. Droplet-sorting would facilitate the segregation of the heterogeneous droplet population as well as manipulation and isolation of the populations of interest. This sorting is accomplished by employing both passive and active methods. Size-dependent sorting relies upon channel geometry that deflects the flow of smaller particles away from the main flow channel [181,182]. Gravity-based sorting, on the other hand, relies on the difference between the sedimentation velocity of larger particles and smaller particles at a given density [183]. Furthermore, the droplets could be precisely actively sorted using dielectrophoresis [184]. Turk-McLeod et al. have recently theorized and implemented a method of ensemble sorting [185]. The method separates groups of objects into a set of registers, which are themselves sorted again, and then re-sorted iteratively (Figure 4). The result is a lossless enrichment process that leads to very high purity and that can cope with errors and highly dynamic populations, leading to high relevance in observation of compartment speciation.

The ability of droplet-based microfluidic devices to run multiple experiments simultaneously in compartmentalized microreactors has, in fact, been translated into many other applications in related fields and topics of study. Droplet-based microfluidics has become an important tool in chemical synthesis, in particular, due to several attractive features: microscale reactions allow for cost reduction through the usage of small reagent volumes, rapid reactions in the order of milliseconds, and efficient heat transfer that leads to environmental benefits when the amount of energy consumed per unit temperature rise can be extremely small [186]. In a microfluidic setup generating temperature gradients, Luccheta et al. have studied *Drosophila melanogaster* embryo development in which temperature perturbations were compensated by a Bicoid morphogen gradient [187]. In cell culture experiments, simultaneous generation of microfluidic droplets of different composition and encapsulation of cells within them allow for high-throughput screening for optimal culture conditions. The isolated environment of each droplet enables the analysis of individual cell populations [188]. Droplet devices are also instrumental in the investigation of conditions necessary for protein crystallization [189,190], while microfluidic implements for the optimization of polymerase chain reaction are being actively developed; several commercial models exist already [191]. In several studies, the directed evolution of enzymes was even optimized utilizing different architectures of a droplet apparatus [192,193,194].

In prebiotic chemistry-related research, Hanczyc utilized a microfluidic device to probe motility in a protocell system [195]. The study showed increased motility of oil droplets impregnated with hydrogen cyanide (HCN) polymer compared to pristine droplets. The active hydrolysis of the HCN polymer within the droplets propelled the droplets by either expelling the reaction products or producing substances altering the surface tension of droplets. Gutierrez et al. [73] investigated the properties of oil microdroplets composed of four different amphiphiles in varying ratios. While not strictly using a microfluidic device, the experimental setup based on a liquid-handling robot that generated the droplets placed them in a Petri dish after which the droplets were recorded using a camera. The behavior of the droplets was analyzed using image recognition software to provide a fitness value. In separate experiments, the fitness function discriminated based on movement, division, and vibration over 21 cycles, giving successive fitness increases. Over the course of the experiment, it appeared that distinct species with different phenotypic properties such as more/less motile, apt to divide, or rotational frequency (or different “species”) emerged at different periods of time. These populations also did not tend to co-exist, rather evolving (as an apparent community) into a different “species” with a different phenotypic property.

These droplet microfluidic devices are obviously highly engineered pieces of equipment that cannot faithfully recreate prebiotic environments. The technique, however, allows researchers to experimentally model prebiotic speciation in laboratory timescales.

## 7. Summary and Conclusions

The concept of species and mechanisms of speciation are highly contested in biology; in prebiotic chemistry, defining species and outlining the processes of speciation is similarly challenging. Herein, we offer a working definition of prebiotic species: delimited chemical systems of similar makeup and properties. The examples of prebiotic speciation include speciation prompted by geographic separation, compositional diversity of macromolecular assemblies, and, most commonly, compartmentalization.

Based on examples of prebiotic compartmentalization studies, we have broken down compartment-driven speciation into two general categories: discriminate and indiscriminate compartmentalization. In the discriminate compartmentalization mode, speciation ensues through a selective partitioning of specific sets of molecules into specific compartments and consequent chemical processes impacted by the selective segregation. Alternatively, the initial homogeneous system can indiscriminately subdivide into compartments. In this case, speciation occurs due to compartmentalization itself altering the reaction condition, e.g., by promoting molecular crowding [148], by creating a low water activity medium [196], or by simply generating multiple initially identical entities later subjected to different environments. In complex chemical systems exhibiting chaotic behavior, one might expect stochastically different outcomes in compartmentalized partitions without an external intervention.

Any conclusion drawn from the studies of biological speciation are based on paleontological data [197], metagenomic studies of environmental niches [67], and artificial speciation [198,199]. In prebiotic chemistry research, observational studies are largely unavailable due both to inaccessibility (to the primitive Earth) and timescale issues (real prebiotic processes likely took significantly longer than what is possible for us to observe directly) as experimental studies are subject to laboratory constraints. While probably unable to precisely reproduce exact prebiotic conditions, droplet microfluidics platforms offer particular advantages in studying compartmentalization-driven speciation. The technique generates in situ droplet compartments that can be either combinatorially composed or impregnated with different loads, resulting in multiple microreactors allowing researchers to probe a myriad of chemical, biochemical, or biological system variants simultaneously. Related techniques had already been utilized to study the chemical evolution of droplet motility [73,195].

The origin of life remains an unsolved scientific question and is a subject of active research and ongoing debate. While we cannot say which set of chemical reactions or conditions ultimately led to emergence of the first living organisms, we can point out benefits to early speciation in protolife systems. The presence of multiple prebiotic species would allow the utilization of various resources, potentially lead to increased stability of each species through compatible interactions, and offer a greater chance of survival in the aftermath of catastrophic events. Prebiotic speciation is, therefore, a topic that warrants further investigation.

## Figures and Tables

**Figure 1 life-11-00154-f001:**
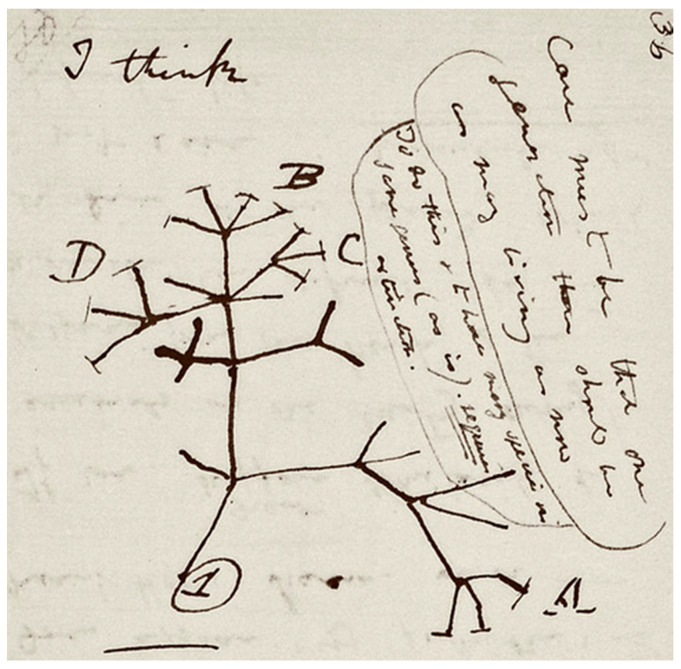
Darwin’s tree of life sketch from his “B” notebook on transmutation of species. Reproduced by kind permission of the Syndics of Cambridge University Library (ms.DaR. 121: p36).

**Figure 2 life-11-00154-f002:**
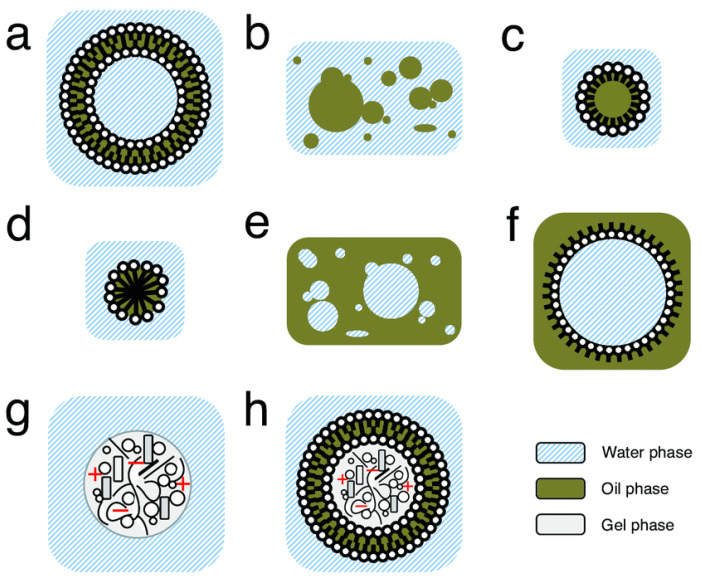
Different structures of potential primitive model compartment systems. (**a**) Unilamellar vesicle. (**b**) Oil droplets in water. (**c**) Surfactant covered oil droplet in water. (**d**) Micelle. (**e**) Water droplets in oil. (**f**) Surfactant covered water droplet in oil. (**g**) Coacervate. (**h**) Vesicle enclosing coacervate. Figure and figure caption adapted and reprinted with permission from ref. [71]. 2016 Shirt-Ediss.

**Figure 3 life-11-00154-f003:**
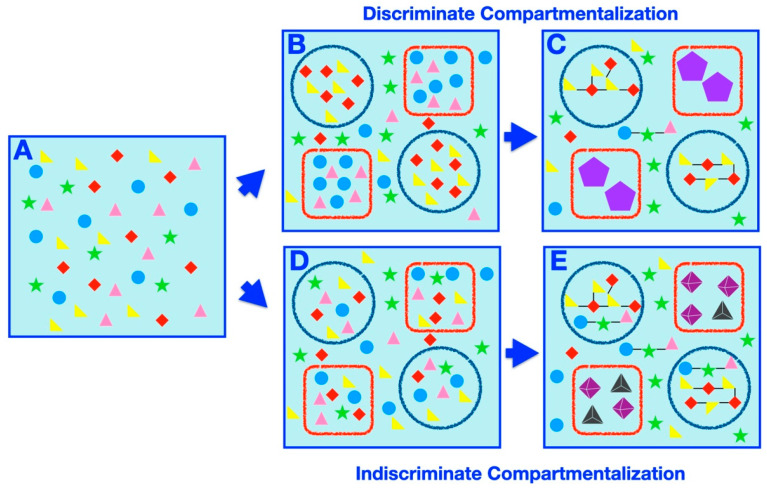
Compartmentalization-prompted speciation: a schematic representation. (**A**) A homogeneous prebiotic chemical mixture (filled shapes of circles, triangles, stars, and diamonds of various colors). (**B**) Two types of discriminating compartments (red square and blue circle) discriminately compartmentalize the chemicals (teal circles and pink triangles in red squares, red diamonds and yellow triangles in blue circles). (**C**) Following further chemical evolution (such as environmental perturbation), this results in distinct compartmentalized macromolecular species (those in the red squares and those in the blue circles). The black lines indicate simple bonds (whether covalent, non-covalent, etc.) formed between the initial chemicals in the mixture (suggestive of the synthesis of larger, and potentially more complex, molecules). The violet pentagons within the red square compartments indicate the formation of complex macromolecular species resulting from significant chemical transformation of members of the initial chemical mixture. In particular, macromolecules in the blue circle compartments are less complex (only containing a few simple bonds connecting the original molecules) than those in the red square compartments (where the initial molecules transformed significantly and contain a number of complex bonds). The identity and composition of the chemicals within each compartment is the main determining factor of the emergence of different compartmentalized macromolecules within different compartments. A number of unencapsulated species may also exist, but they are distinct in structure and character from the encapsulated species. (**D**) Two types of compartments (red square and blue circle) indiscriminately compartmentalize the chemicals at random, with each type of chemical residing in every compartment. (**E**) Following further chemical evolution (such as environmental perturbation), this results in distinct compartmentalized species (those in the red squares and those in the blue circles), where the properties and structure compartment itself contributes significantly to the formation of different encapsulated small molecules and macromolecules within different compartments (more so than the identity or composition of the encapsulated chemicals, which is identical in each type of compartment). The black lines indicate simple bonds (whether covalent, non-covalent, etc.) formed between the initial chemicals in the mixture (suggestive of the synthesis of larger, and potentially more complex, macromolecules). The tetrahedra and octahedra within the red square compartments indicate the formation of complex macromolecular species resulting from significant transformation of members of the initial chemical mixture. In particular, macromolecules in the blue circle compartments are less complex (only containing a few simple bonds connecting the original molecules) than those in the red square compartments (where the initial molecules transformed significantly and contain both a number of complex bonds and supramolecular self-assembly characteristics). A number of unencapsulated species may also exist, but they are distinct in structure and character from the encapsulated species.

**Figure 4 life-11-00154-f004:**
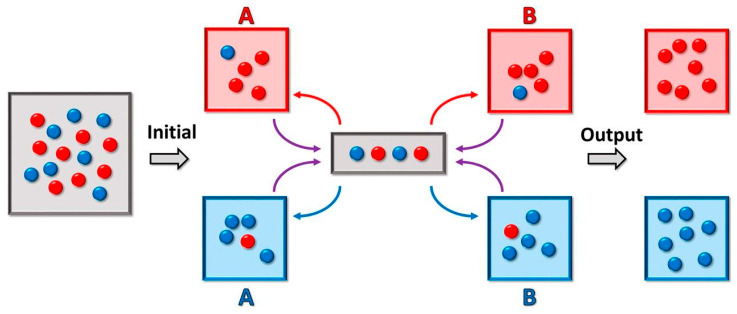
Microfluidics-based droplet ensemble sorting method from [185]. The sorter contains four different “registers”, red and blue for both (A) and (B). The gray area represents the measurement area, which results in sorting of each droplet within that area into one of the four registers. Iterative sorting between the (A) and (B) registers is applied until all red droplets reside in the same red register, and all blue droplets reside in the same blue register. Figure reprinted with permission from ref. [185]. 2018 PNAS.

**Table 1 life-11-00154-t001:** Glossary of terms discussed in this paper.

Term	Definition
Allele	A version of a gene which may contain variations different from other versions of the same gene. In some biological systems, more than one gene variant exists (for example, two of each chromosome exists in biploid organisms like humans, while multiple forms of each chromosome exist in polyploid organisms like wheat). An allele is simply one of these gene variants.
Allopatry or Allopatric Speciation	Speciation resulting from geographical isolation of two populations of the same original species.
Autotrophic	A process by which a system obtains energy that drives its own function (such as replication, growth, evolution, etc.) directly from an inorganic source (such as minerals, geological processes, sun, etc.).
Autotrophic Origin of Life	A theory assuming that life originated from geochemical processes patterned after non-enzymatic metabolic reactions, most commonly, the reverse Krebs cycle.
Biological Concept of Species (BCS)	Groups of actually or potentially interbreeding natural populations, which are reproductively isolated from other such groups.
Biological Speciation	The process by which biological lineages diverge from their ancestors to become new biological species as a result of differential selective pressures, geographical isolation, or other processes.
Chemical Speciation	The process by which delimited chemical systems of similar makeup and properties are subjected to differential environments or chemical/physical conditions, resulting in emergence of more than one distinct delimited chemical system with different makeup and properties from each other.
Coacervates	Macromolecule-rich dispersed droplets in liquid–liquid phase separated systems. According to Oparin and Haldane, coacervates constitute protocells.
Compartmentalization	The physical process by which an analyte stably resides within a compartment for long periods of time and does not rapidly depart from that compartment on short timescales.
Complex Coacervates	A coacervate, but specifically one resulting from phase separation caused by electrostatic binding of oppositely charged polyions.
Composome	The compositional information of a collection of molecules before the emergence of life, often referred to as part of the graded autocatalysis replication domain (GARD) model, which assumes that primitive functions such as replication and catalysis could have existed before the emergence of replicating polymers.
Discriminate Compartmentalization	The process by which specific subsets of molecules are placed within specific compartments due to systematic factors (such as physicochemical properties of the molecules and/or the compartments).
Ecological Niches	The placement of an organism within a specific ecological community, and the various interactions this organism has with the environment and other members of the community.
GARD (Graded Autocatalysis Replication Domain)	A general kinetic model in evolutionary biology for homeostatic growth and fission of compositional assemblies with specific application toward lipids.
Heterotrophic	A process by which a system obtains energy that drives its own function (such as replication, growth, evolution, etc.) from other non-inorganic sources (such as other chemicals, polymers, organisms, etc.).
Heterotrophic Origin of Life	A theory assuming that early organisms depended on abiotically synthesized organic molecules for their structural components and as an energy source.
Homologous Recombination	Genetic information exchange occurring between two identical (or nearly identical) gene-carrying forms (alleles). For example, this would occur between two different alleles of homologous (identical) genes between similar organisms, akin to sexual reproduction.
Horizontal Gene Transfer	The transfer of genetic information not acquired directly from a parent but rather from other members of the same (or similar) community of organisms that may come from the same lineage.
Indiscriminate Compartmentalization	The process by which random samplings of molecules are partitioned within compartments arbitrarily, independent of the physicochemical properties of the molecules or the compartments.
Liquid–liquid Phase Separation (LLPS)	The process by which a mixture two liquids separate from each other (due to thermodynamic reasons such as immiscibility, differences in chemical affinities, entropic effects, etc.), resulting in two immiscible liquids co-existing.
LUCA (Last Common Universal Ancestor)	The earliest single-celled form of life on Earth which eventually evolved into all known forms of modern life.
Membrane Liposome/Vesicle	A specific compartment type that has been proposed as a protocell and primitive compartment model.
Messy Chemistry	A field of prebiotic chemistry which assumes that the prebiotic milieu is sufficiently complex (i.e., “messy”), and whose experiments focus on bulk properties of complex prebiotic chemical mixtures (such as emergent functions or reactions) as opposed to characterizing exact mechanisms of primitive processes (it is impossible to study such processes on the actual prebiotic Earth).
Nonhomologous Recombination	Genetic information exchange which occurs as a result of ligation of two (or more) essentially unrelated genes or nucleic acids.
Parapatry or Parapatric Speciation	Speciation resulting from partial geographical separation afforded to diverging populations.
Peripatry or Peripatric Speciation	A form of allopatric speciation in which a small population is isolated on the “periphery” of a large population, and the reduced population eventually can no longer breed with the main population.
Phenotypic	An observable or expressed trait in any organism, species, system, etc., that may be governed by its genetic, chemical, structural, etc. composition.
Primitive Compartment	A compartment that can be assembled using prebiotically available building blocks on early Earth; this compartment need not have directly evolved into modern cells and may have provided other functions to primitive chemical systems.
Progenote	A proposed form that early life on Earth could have taken on. Rather than a single-celled LUCA, a progenote was a collection of cells which communicated and evolved through horizontal transfer of encapsulated components (i.e., genetic/catalytic biopolymers).
Protocell	A primitive compartment that necessarily evolved directly into modern cells. A protocell is a specific type of primitive compartment, but a primitive compartment is not necessarily a protocell.
Protolife Systems	These are systems which emerged before the origin of life and necessarily evolved into modern life. They may have been composed of protocells, or of a collection of protocells, or simply of a collection of chemicals.
Random Genetic Drift	Random fluctuations in the frequency of expression of genes within a population (such as variation of alleles) resulting from random statistical occurrences (such as sudden reduction of population size resulting in more variation of gene expression).
Sympatry or Sympatric Speciation	Speciation resulting from an absence of geographical separation of any form.
Taxonomy	The study of classifying organisms based on (generally) patterns of their genetic inheritance. Taxonomy can also be applied to classifying chemical systems based on chemical properties or structures.
Vertical Transmission	The transfer of genetic information directly from a parent organism to an offspring organism.

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
