# Peer review of "Origin of Species before Origin of Life: The Role of Speciation in Chemical Evolution"

_life, 2021, doi:10.3390/life11020154_

Round 1

Reviewer 1 Report

This review focuses on the question of the emergence of speciation in earth-life history. The authors discuss the mechanisms of speciation in the context of biology, and then shift focus to origins of life and chemical evolution. They focus on the compartment emergence before LUCA, and other mechanisms underlying speciation prior to LUCA, as well as primitive compartment structures, and modes of such compartmentalization. The authors specifically introduce droplet microfluidic technology to study speciation in chemical systems.

The authors tackle a big problem in biology, and I command them for articulating this problem in the context of the origins of life (even harder). This is a complex problem. Yet they synthesize the literature very well, both on the biological and chemical end, and present a coherent picture. The article is easy to read, and pedagogical. I learned a lot from reading it, I thank the author for their contribution. I have a few minor comments to improve the flow, and some suggestions I outline them line by line below, and I have a few other questions regarding the philosophy behind their question. Other than these, article is in good shape and should be accepted after these changes.

My main request is for the authors to better justify their choice of the technological emphasis: microfluidics, as a methods solution towards understanding the speciation in early chemical systems. I believe they discuss the technology and recent developments fairly well, but I believe the flow towards the transition into the microfluidcs technology is weak. Why this method over others? Perhaps this information is present in the review, but the intensity of the methodological text (fairly deep technological description, but takes the reader away from the main topic in my opinion) -- makes it difficult to extract the information regarding: why microfluidics. 

Secondly, I think the paper would benefit a Glossary table where all the new/used technical/discipline specific terms are outlined. I specifically would like to see the authors define the difference between a protocell vs. membrane liposomes. I think these days anyone who is able to generate a membrane in the laboratory refers to these lipids as “protocells”. Given that the senior author of the paper is a leader in the field, and that the paper focuses on this very topic, I think this is an excellent place to define the difference between these two for once and for all! Other terms in the Glossary table should include: discriminate, indiscriminate, microdroplets, speciation (!!) — (biological and chemical/prebiotic), messy chemistry, compartmentalization, protolife systems, etc.

Along these lines, authors define prebiotic species as: delimited chemical systems of similar makeup and properties. Such definition does not provide insights into mechanism, I believe the definition needs to be deepened to include the mechanism of speciation formation as well as to define the “make up and properties”. What do authors mean by that?

Missing citations (Im not author on these):

Vincent and Baum’s recent work chemical gardens (Life, 2019)

Joseph Moran’s recent work on chemical networks (Chem reviews, 2020)

Adam, Aono and Fahrenbach’s work on prebiotic chemical automata at the origins of life (Complexity, 2019)

Mayr’s what is a species (1996) - in addition to the other two they cited

Other minor comments:

Page 1: “give or take” — too colloquial, please edit

Page 3, needs a smoother transition (perhaps a sentence or two at the end of the paragraph above part 2),  summarizing the “big message of the paper”.

Citation 26 is unformatted.

Page 4

Because authors present Darwin’s famous tree, can they also present such a tree for lipid membranes? It would be a nice figure side by side and would emphasize the protocell/lipid and chemical evolution point of the paper from a historical view.

Page 5

Here cite Adam and Aono, who mapped out stratified relationships between physical components that drive the production of key prebiotic molecules, as well as as Baum's fairly recent work published in Life (see above).

Page 6

“In order to accomplish these functions on early Earth, a primitive compartment would have required some ..” — because this is title sentence, clarify what you mean by “these functions”

Figure 2: text size is inconsistent (c is smaller than b)

Page 7

GARD model — this comes out of nowhere (as an unfamiliar reader on the topic) even though mentioned once before a couple pages above, clarify again.

Page 8:

“In both cases, a significant divergence in the structure (and potentially function as well) of those entities within and on the outside of the compartment would have resulted.”

This sentence is unclear.

“Gradually, the population of the selected species will grow and get transformed in turn, eventually resulting in compartment-level speciation” — this needs a citation

Page 9, Figure 3: annotations need to be clarified, the black lines in C are not clear (what do they indicate?) — also, please mind the color blind (red/green is hard to tell apart)

Author Response

This review focuses on the question of the emergence of speciation in earth-life history. The authors discuss the mechanisms of speciation in the context of biology, and then shift focus to origins of life and chemical evolution. They focus on the compartment emergence before LUCA, and other mechanisms underlying speciation prior to LUCA, as well as primitive compartment structures, and modes of such compartmentalization. The authors specifically introduce droplet microfluidic technology to study speciation in chemical systems.

The authors tackle a big problem in biology, and I command them for articulating this problem in the context of the origins of life (even harder). This is a complex problem. Yet they synthesize the literature very well, both on the biological and chemical end, and present a coherent picture. The article is easy to read, and pedagogical. I learned a lot from reading it, I thank the author for their contribution. I have a few minor comments to improve the flow, and some suggestions I outline them line by line below, and I have a few other questions regarding the philosophy behind their question. Other than these, article is in good shape and should be accepted after these changes.

We thank the reviewer for their careful review of our manuscript and for their comments and suggestions, which we believe have significantly improved the quality of this manuscript.

My main request is for the authors to better justify their choice of the technological emphasis: microfluidics, as a methods solution towards understanding the speciation in early chemical systems. I believe they discuss the technology and recent developments fairly well, but I believe the flow towards the transition into the microfluidcs technology is weak. Why this method over others? Perhaps this information is present in the review, but the intensity of the methodological text (fairly deep technological description, but takes the reader away from the main topic in my opinion) -- makes it difficult to extract the information regarding: why microfluidics. 

We have added clarifications to better frame the section in the context of the overarching theme of the paper by adding a few more lines of introduction which further expounds upon the relevance of utilizing microfluidics technologies to study speciation phenomena of compartments in the lab. We have also slightly decreased the amount of detailed technical information as a means to promote easier information extraction and readability.

Secondly, I think the paper would benefit a Glossary table where all the new/used technical/discipline specific terms are outlined. I specifically would like to see the authors define the difference between a protocell vs. membrane liposomes. I think these days anyone who is able to generate a membrane in the laboratory refers to these lipids as “protocells”. Given that the senior author of the paper is a leader in the field, and that the paper focuses on this very topic, I think this is an excellent place to define the difference between these two for once and for all! Other terms in the Glossary table should include: discriminate, indiscriminate, microdroplets, speciation (!!) — (biological and chemical/prebiotic), messy chemistry, compartmentalization, protolife systems, etc.

This is a really great suggestion, and we have included a glossary of terms in the introduction as Table 1. 

Along these lines, authors define prebiotic species as: delimited chemical systems of similar makeup and properties. Such definition does not provide insights into mechanism, I believe the definition needs to be deepened to include the mechanism of speciation formation as well as to define the “make up and properties”. What do authors mean by that?

We have included further clarification of the “make up and properties” of chemical systems, which include their structure, reactivity, and/or other physicochemical properties such as dissociation temperature.

Missing citations (Im not author on these):

Vincent and Baum’s recent work chemical gardens (Life, 2019)

Joseph Moran’s recent work on chemical networks (Chem reviews, 2020)

Adam, Aono and Fahrenbach’s work on prebiotic chemical automata at the origins of life (Complexity, 2019)

Mayr’s what is a species (1996) - in addition to the other two they cited

We have described the relevant studies and cited each of these studies throughout the article in the appropriate locations 

Other minor comments:

Page 1: “give or take” — too colloquial, please edit

We have changed this phrase to “with an error margin of plus or minus”.

Page 3, needs a smoother transition (perhaps a sentence or two at the end of the paragraph above part 2),  summarizing the “big message of the paper”.

We have added some additional statements at the end of the introduction which summarize the “big message” of the paper, which focuses on discussing both that the origin of species preceded the origin of life as well as the need for further studies and understanding that combine understanding of speciation in the context of biology with theoretical/experimental research on primitive compartments.

Citation 26 is unformatted.

We believe that it is formatted correctly. However, in the text, we have moved this citation to the end of the sentence to minimize any potential confusion.

Page 4

Because authors present Darwin’s famous tree, can they also present such a tree for lipid membranes? It would be a nice figure side by side and would emphasize the protocell/lipid and chemical evolution point of the paper from a historical view.

We thank the reviewer for this suggestion, but believe that a discussion on lipid membrane evolution at this point of the paper would not be relevant or appropriate (and beyond the scope of the paper in general as well). In particular, the section where this figure is presented focuses on the concept of biological speciation, and the fact that the Darwin tree is essentially the root of this entire concept is the main reason that it was presented. We believe that this seminal work is of utmost historical significance, and in comparison, a lipid membrane evolutionary tree of similar significance does not exist. As such, we have decided not to include such a figure. However, there is much interesting work on the possibilities as to how lipids of Archaea and Eukarya initially speciated from LUCA, although there are still no definitive answers. This study discusses some of the possibilities, and is worth a read: Shin-ichi Yokobori, Yoshiki Nakajima, Satoshi Akanuma, Akihiko Yamagishi. "Birth of Archaeal Cells: Molecular Phylogenetic Analyses of G1P Dehydrogenase, G3P Dehydrogenases, and Glycerol Kinase Suggest Derived Features of Archaeal Membranes Having G1P Polar Lipids", Archaea, vol. 2016, Article ID 1802675, 2016. https://doi.org/10.1155/2016/1802675

Page 5

Here cite Adam and Aono, who mapped out stratified relationships between physical components that drive the production of key prebiotic molecules, as well as as Baum's fairly recent work published in Life (see above).

We have cited and briefly summarized these studies.

Page 6

“In order to accomplish these functions on early Earth, a primitive compartment would have required some ..” — because this is title sentence, clarify what you mean by “these functions”

We have clarified “these functions” accordingly in the text.

Figure 2: text size is inconsistent (c is smaller than b)

Although we believe that the font size in the figure was consistent, we noticed that the original figure text placement was not consistent (for example, c was a number of pixels lower than a and b), which may have led to this observation. As such, we have fixed this inconsistency in the figure and modified the figure caption to clearly show that part of the figure was adapted.

Page 7

GARD model — this comes out of nowhere (as an unfamiliar reader on the topic) even though mentioned once before a couple pages above, clarify again.

We have included some more brief description about the specifics of the GARD model to remind readers of the information presented above.

Page 8:

“In both cases, a significant divergence in the structure (and potentially function as well) of those entities within and on the outside of the compartment would have resulted.”

This sentence is unclear.

 We have clarified this sentence in the text to be more specific as to what “cases” and “entities” refers to.

“Gradually, the population of the selected species will grow and get transformed in turn, eventually resulting in compartment-level speciation” — this needs a citation

We have now cited two studies, which show that (1) selective advantages such as growth can be imbued to a compartment system through function of catalytic molecules (such as ribozymes) within a compartment [Adamala, K.; Szostak, J.W. Competition between model protocells driven by an encapsulated catalyst. Nat. Chem. 2013, 5, 495–501.], and that (2) faster growth can also be imbued to a compartment system through more efficient replication of encapsulated polymers, which would drive Darwinian evolution, and subsequently speciation, at the protocellular (compartment) level [Chen, I.A.; Roberts, R.W.; Szostak, J.W. The emergence of competition between model protocells. Science 2004, 305, 1474–1476.].

Page 9, Figure 3: annotations need to be clarified, the black lines in C are not clear (what do they indicate?) — also, please mind the color blind (red/green is hard to tell apart)

We have added annotations to the figure legends that we hope are more descriptive of the concepts in the figure we hoped to convey. We also believe that the shapes can help to distinguish the different chemicals in addition to color. To further improve understandability, we have finally combined figures 3 and 4 to improve the ability for the reader to compare both effects.

Reviewer 2 Report

Dear editor, dear authors,

the manuscript "Origin of Species Before Origin of Life: The Role of Speciation in Chemical Evolution" submitted to Life represents an excellent review which i fully recommend for publication in the journal Life.

For improvement, I only have the following minor comments:

# line 55 (and later)
Wachtershauser -> Wächstershäuser (ä instead of a)

# line 120-121, line 145-146
On the Origin of the Species -> "On the Origin of the Species" (add parantheses)

# figure 3 / figure 4
I suggest to combine both figures into one which might make it easier for the reader to compare both effects.

Author Response

Dear editor, dear authors,

the manuscript "Origin of Species Before Origin of Life: The Role of Speciation in Chemical Evolution" submitted to Life represents an excellent review which i fully recommend for publication in the journal Life.

We thank the reviewer for their careful review of our manuscript and for their comments and suggestions, which we believe have improved the quality of this manuscript.

For improvement, I only have the following minor comments:

# line 55 (and later)

Wachtershauser -> Wächstershäuser (ä instead of a)

We have incorporated this change throughout the text.

# line 120-121, line 145-146

On the Origin of the Species -> "On the Origin of the Species" (add parantheses)

We have incorporated this change and represent this and other titles in Italics within double-quotes.

# figure 3 / figure 4

I suggest to combine both figures into one which might make it easier for the reader to compare both effects.

We have combined both figures and hope that the new format is easier to understand and compare both effects.